# Linear Diagnostic Procedure Elicited by Clinical Genetics and Validated by mRNA Analysis in Neuronal Ceroid Lipofuscinosis 7 Associated with a Novel Non-Canonical Splice Site Variant in *MFSD8*

**DOI:** 10.3390/genes14020245

**Published:** 2023-01-17

**Authors:** Domizia Pasquetti, Giuseppe Marangi, Daniela Orteschi, Marina Carapelle, Federica Francesca L’Erario, Romina Venditti, Sabrina Maietta, Domenica Immacolata Battaglia, Ilaria Contaldo, Chiara Veredice, Marcella Zollino

**Affiliations:** 1Unit of Medical Genetics, Section of Genomic Medicine, Department of Life Sciences and Public Health, Università Cattolica del Sacro Cuore, Fondazione Policlinico Universitario A. Gemelli IRCCS, 00168 Rome, Italy; 2Child Neurology and Psychiatry Unit, Department of Neuroscience, Catholic University of the Sacred Heart, Fondazione Policlinico Universitario Agostino Gemelli IRCCS, 00168 Rome, Italy

**Keywords:** neurodevelopmental disorders, clinical genetics, *MFSD8*, CNL7, mRNA

## Abstract

Neuronal ceroid lipofuscinoses (CNL) are lysosomal storage diseases that represent the most common cause of dementia in children. To date, 13 autosomal recessive (AR) and 1 autosomal dominant (AD) gene have been characterized. Biallelic variants in *MFSD8* cause CLN7 type, with nearly 50 pathogenic variants, mainly truncating and missense, reported so far. Splice site variants require functional validation. We detected a novel homozygous non-canonical splice-site variant in *MFSD8* in a 5-year-old girl who presented with progressive neurocognitive impairment and microcephaly. The diagnostic procedure was elicited by clinical genetics first, and then confirmed by cDNA sequencing and brain imaging. Inferred by the common geographic origin of the parents, an autosomal recessive inheritance was hypothesized, and SNP-array was performed as the first-line genetic test. Only three AR genes lying within the observed 24 Mb regions of homozygosity were consistent with the clinical phenotype, including *EXOSC9*, *SPATA5* and *MFSD8*. The cerebral and cerebellar atrophy detected in the meantime by MRI, along with the suspicion of accumulation of ceroid lipopigment in neurons, prompted us to perform targeted *MFSD8* sequencing. Following the detection of a splice site variant of uncertain significance, skipping of exon 8 was demonstrated by cDNA sequencing, and the variant was redefined as pathogenic.

## 1. Introduction

Neuronal ceroid lipofuscinoses (NCLs) are a group of inherited neurodegenerative lysosomal storage diseases that together represent the most common cause of dementia in children [1]. Despite clinical and genetic heterogeneity, all forms of NCLs are characterized by an accumulation of autofluorescent lipopigments in different cell types and by a loss of neuronal cells in the brain and, in most forms, retina [2]. These lipopigments show distinct ultrastructural patterns, i.e., granular, curvilinear/rectilinear and fingerprint profiles [3].

Historically, classification has been based on age of onset together with clinical presentation, but, over time, several so-called variant forms have been recognized. Therefore, according to the presenting phenotype, patients were originally clustered into the six different NCL subtypes [4], consisting of congenital (CLN10), infantile (CLN1), late infantile (CLN2), variant late infantile (CLN5, CLN6, CLN7 and CLN8), juvenile (CLN3) and adult (CLN4) NCL. Among those, the late-infantile subtype is the most frequent and it is characterized by great genetic variability, with pathogenic variants in at least eight known genes. The identification of the genetic defects that underlie the majority of NCL cases led to an updated NCL nomenclature, which is genetically based, but still takes into account key clinical and pathological features. In the latter nomenclature, the specific subtype of the disease is defined by the causative gene, followed by the clinical presentation features (for example, CLN1 disease, late infantile). 

As an accumulation storage group of disorders, the natural history of NCL is typically progressive and exhibits cognitive and psychomotor regression with seizures, visual impairment, ataxia, and myoclonus [5].

To date, 13 autosomal recessive and one autosomal dominant gene have been associated with NCL, including both secreted lysosomal protein-coding genes (CLN1, CLN2, CLN5, CLN10 and CLN13) and transmembrane protein-coding genes (CLN3, CLN4, CLN6, CLN7, CLN8, CLN12 and CLN14) [6].

CLN7 (OMIM #610951) is caused by biallelic pathogenic variants in the *MFSD8* gene, which encodes a protein of major facilitator superfamily (MFS). It is a 518 amino acid lysosomal transmembrane protein that appears to be involved in the translocation of small solutes across the membrane [7]. However, the exact nature of metabolites transported by the MFSD8 protein and the pathogenesis of the progressive neurodegeneration are still largely unknown [8]. Patients usually exhibit an infantile onset (between two and seven years of life) of classic NCL symptoms, including seizures, visual loss, mental regression and ataxia [9]. Hypomorphic *MFSD8* variants have been reported as causative of a nonsyndromic form with only macular disease [10], or widespread rod-cone dystrophy with severe macular involvement [11].

Since the initial identification of the causative gene in 2007, more than 50 different *MFSD8* variants responsible for CLN7 have been reported in the literature [7,9,12,13,14,15,16,17,18,19,20,21,22,23,24,25,26,27,28,29,30] and a comprehensive list can be found in the “NCL Mutation and Patient Database” by the University College of London (UCL) [https://www.ucl.ac.uk/ncl-disease/mutation-and-patient-database (accessed on 5 January 2023)]. Most patients were diagnosed with missense and stop-gain variants; however, canonical splice site variants, single exon or whole exon deletions were also identified in a few.

We report here clinical and molecular findings of a 5-year-old girl who presented with developmental regression and microcephaly, in whom a novel homozygous intronic variant in *MFSD8*, inherited from healthy parents, was detected by targeted gene sequencing on genomic DNA first, and then validated by cDNA sequencing. In describing the diagnostic procedure, we aim to highlight the pivotal role of clinical genetics, integrated with a multidisciplinary assessment and mRNA analysis, in reaching a linear and unequivocal diagnosis of CLN7. We also made an extensive literature revision regarding both clinical and molecular features of published CLN7 patients.

## 2. Materials and Methods

### 2.1. Patient Study and Literature Review

A 5-year-old girl was admitted to the Child Neurology Department at Gemelli Hospital in Rome because of unexplained progressive neurodevelopment impairment and microcephaly. She immediately underwent a multidisciplinary assessment, including neuroimaging, neuro-cognitive scale grading, metabolic disease screening and clinical genetic evaluation. On the clinical evidence of a monofactorial genetic condition, genetic investigations were planned after obtaining written informed consent from both parents. Clinical and genetic data of our patient were comparatively evaluated with those of patients reported in the literature (tot 76, see Appendix A) [7,9,12,13,14,15,16,17,18,19,20,21,22,23,24,25,26,27,28,29,30] and collected in the UCL NCL database cited above (compared to the database, we only considered patients with ascertained neurological involvement clinically described in the scientific literature, thus excluding personal communications and patients in whom there was only isolated retinal disease).

### 2.2. Analyses on Genomic DNA

We performed chromosomal microarray analysis with the commercial GenetiSure Cyto CGH + SNP Microarray kit (Agilent Technologies, Santa Clara, CA, USA), following manufacturer instructions, using the ADM-2 algorithm for data analysis with Agilent CytoGenomics software.

After evaluating neuroimaging together with the gene content of the homozygous regions detected by SNP-array, direct sequencing of all coding exons and exon–intron junctions of the *MFSD8* gene (“MANE Select” transcript NM_001371596.2) was performed. Primer sequences are listed in Appendix A. The Sanger sequencing reaction was prepared with the BigDye™ Terminator v3.1 Cycle Sequencing Kit (ThermoFisher Scientific, Waltham, MA, USA), purified with BigDye XTerminator™ Purification Kit (ThermoFisher Scientific), run on an Applied Biosystems 3130xl Genetic Analyzer (ThermoFisher Scientific) and analysed with Sequencing Analysis Software v6.0 (ThermoFisher Scientific), according to manufacturer instructions.

Variant inheritance was assessed on genomic DNA from both parents by direct sequencing of the same regions.

Sequence variants were described according to the Human Genome Variation Society nomenclature guidelines (https://varnomen.hgvs.org/ (accessed on 12 January 2023)).

### 2.3. Analyses on cDNA

Short-term PHA-stimulated lymphocyte cultures were established from the patient’s heparinized blood samples following standard procedures [31]. Total RNA was hence extracted from both untreated and puromycin-treated cells using the guanidinium thiocyanate–phenol–chloroform extraction method [32]. We used the RNA sample extracted using the same procedure from an, at least apparently, healthy adult individual.

cDNA was obtained by reverse transcription of RNA with the High-Capacity cDNA Reverse Transcription Kit (ThermoFisher Scientific). Target amplification of *MFSD8* transcript was obtained with standard PCR with primer pairs overlapping the junction between exons 6 and 7 (NM_001371596.2, primer sequence: CTTGCCATACTAAGAGAACATCG) and exon 10 (CAGTAGAATAGCACGCTCGC). Sanger sequencing reaction of the amplified cDNA segments was eventually performed on RNA samples from both the patient and a control subject for comparison.

## 3. Results

### 3.1. Clinical Report

This is a 5-year-old female, second-born to healthy and apparently unrelated parents of Italian ancestry. The same geographic origin from a village in Sicily was recorded for the parents. Family history was not informative for any known genetic diseases; she has a healthy older brother. She was born at 38 weeks by normal delivery after an uneventful pregnancy. Prenatal ultrasounds were repeatedly normal. Birth weight was 3500 g (75th–90th centile), length was 50 cm (75th centile) and head circumference was 34 cm (50th–75th centile). Neonatal period was uneventful.

She was first referred to the child neurologist at the age of 12 months due to sporadic episodes of tremor at the left lower limb. EEG and neurologic evaluation provided normal results at this time. At the age of 3 years, she experienced progressive regression of motor skills with sub-acute onset of instability, balance impairment, bradykinesia and partial loss of normal gait. Language development was delayed: she was able to speak her first words at 12 months of age, followed by sudden developmental stagnation. As reported by parents, she also developed mild behavioral abnormalities, including poor interest in peers and occasional motor stereotypes. The worsening clinical picture required hospitalization in our Child Neurology Department. In the acute setting, a CT brain scan excluded neurological emergencies. The clinical phenotype consisted of psychomotor delay, mild cognitive impairment, bradykinesia, disturbance of balance and walking, drop attacks and limbs and trunk myoclonic jerks. Brain and spinal cord MRI showed diffuse and bilateral cerebellar atrophy and bilateral and symmetrical hyperintensity of the posterior periventricular white matter and mesial portions of the thalami on T2 and FLAIR sequences (Figure 1A,B). The awake and sleep video-EEG showed focal and diffuse epileptiform discharges on a diffuse slow background activity (Figure 1C), which confirmed the epileptic origin of the myoclonic jerks. It also highlighted frequent atypical absences. Metabolic analyses of blood and urine gave normal results. Structural assessment of neurodevelopmental impairment, through the Leiter III scale, revealed a moderate intellectual disability (ID) (QIT: 54). On ophthalmologic evaluation, a significantly impaired visual acuity was observed without any identifiable reason. Only after the molecular diagnosis were more accurate investigations suggested (i.e., assessment of the ocular fundus and optical coherence tomography, OCT). On cardiac evaluation, neither echocardiographic nor electric alterations were detected. 

At our first genetic counselling, at the age of 4^7/12^ years, true microcephaly was noted, with weight of 15 kg (50th centile), height of 104 cm (50th centile) and head circumference of 46.8 cm (−3 DS). True facial dysmorphims were absent, however some peculiar characteristics were noted, including rounded nasal tip, thin nasal bridge and deep-set eyes.

### 3.2. Genetic Testing

Array-CGH/SNP analysis identified two large regions of homozygosity on chromosome 4 (19.2 Mb) and chromosome 12 (5.3 Mb), respectively, (arr[GRCh38] 4q27q31.1(120508690_139771691)x2 hmz, 12q12q13.12(43904747_49239943)x2 hmz). A phenotype-oriented evaluation of the gene content of both regions led to the selection of the *MFSD8* gene (OMIM: * 611124, responsible for neuronal ceroid lipofuscinosis, type 7) for further analyses.

Sanger sequencing of *MFSD8* allowed for the detection of the homozygous intronic variant NC_000004.12(NM_001371596.2):c.863 + 2dup (Figure 2). The variant does not affect the canonical donor splice site “GT” but leads to a one-base shift in the downstream sequence and to subsequent change in the donor site 5th base (from G to A). SpliceAI algorithm (https://spliceailookup.broadinstitute.org/ (accessed on 5 December 2022)) predicted a high chance for donor site loss, returning a score of 0.92, in a range from 0 to 1. The variant is not present in either the 2.1.1. or 3.1.2 versions of the gnomAD database (https://gnomad.broadinstitute.org/ (accessed on 5 December 2022)). Both parents were heterozygous carriers for the same variant (Figure 2). 

Since relevant consequences to the physiological splicing of *MFSD8* mRNA are expected, with major changes to the transcript sequence (such as, for instance, the skipping of an exon or the retention of intronic segments), we performed cDNA sequencing to investigate this aspect. We could only detect transcript fragments lacking the whole of exon 8 (Figure 2). Indeed, sequencing results showed the presence, in comparable amounts, of a fraction of transcripts lacking the whole of exon 9, as well. Both RNA samples derived from puromycin-treated and untreated cells displayed the same results. *MFSD8*-cDNA Sanger sequencing in a control uniquely detected fragments containing both exons 8 and 9, as expected. Based on these results, we could infer that the variant identified is responsible for the skipping of exon 8 and prevents the synthesis of the normal NM_001371596.2 transcript. A truncated protein may be translated (NP_001358525.1: p.(Thr254Leufs*4)) from the observed transcript, since at least some mRNA molecules seem to escape nonsense-mediated decay. We cannot establish if the fragment lacking exon 9 is likewise a consequence of the variant effect or the product of an alternative transcript that is physiologically present at very low levels in cells and that became detectable only because of the absence of the normal transcript.

## 4. Discussion

We share here the diagnostic procedure by which an unequivocal diagnosis of *MFSD8*-associated CLN7 was reached by targeted gene sequencing in a 5-year-old girl presenting with unexplained developmental regression, progressive microcephaly of postnatal onset and no facial dysmorphisms. The diagnostic procedure was elicited by clinical genetics, with the support of a multidisciplinary approach, which included metabolic investigations, neurological evaluation and brain MRI. 

First, the clinical phenotype was considered highly consistent with a monogenic, although nonspecific, condition. Parental consanguinity was not reported; however, the same geographic origin of the parents prompted us to hypothesize an autosomal recessive (AR) inheritance of the condition in the daughter, and SNP-array was performed as the first-line genetic test, with the main purpose to look for regions of homozygosity. Based on the preliminary diagnosis of cerebral and cerebellar atrophy on brain MRI, only three genes causative of AR forms of white matter diseases were lying within the observed 24 Mb regions of homozygosity, including *EXOSC9* (OMIM *606180)*, SPATA5* (OMIM *613940) and *MFSD8* (OMIM *611124). Therefore, they were tentatively considered strong candidate genes. Since, in retrospect, MRI imaging was consistent with a form of NCL, we decided to perform targeted *MFSD8* sequencing on genomic DNA. A non-canonical splice site variant of uncertain significance in intron 8 was detected. Skipping of exon 8 was then demonstrated by cDNA sequencing, and the variant was redefined as pathogenic.

Regarding the genomic defect of the reported *CLN7* patients from both the scientific literature [7,9,12,13,14,15,16,17,18,19,20,21,22,23,24,25,26,27,28,29,30] and the UCL NCL database, missense and stop-gain variants (including both frameshift and nonsense) account for 44% and 27% of the total mutations, respectively. Missense variants are distributed across the whole length of the protein without any hotspot. At least some of them may act, from a functional point of view, as null mutations: in fact, Steenhuis and colleagues [33] demonstrated that p.Thr294Lys and p.Pro412Leu lead to enhanced proteolytic cleavage of mutant MFSD8 by cysteine proteases, inactivating the protein. In addition, some missense variants, including the p.Thr294Lys itself, are recurrent in different case studies [13]. Partial [29] or whole gene [30] deletions are uncommon in *CLN7*, while splice site defects represent 21% of the different kinds of variants. 

Interestingly, the majority of known splice site variants affect the donor site in intron 7 and 8 of the NM_001371596.2, which is the MANE Select transcript we use as reference. Instead, they correspond to exons 8 and 9 of the NM_152778.1 transcript that was considered in other studies and that has the same coding sequence. The c.754 + 2T > A variant was reported in four patients and was demonstrated to result in an almost complete lack of the normal transcript and an increased expression of two alternatively spliced transcripts: one lacking exon 7 and one lacking exons 6 and 7 [7]. The exonic variant c.750A > G, apparently synonymous, was demonstrated to significantly reduce the expression of the full-length mRNA and increase the transcription of an mRNA skipping exon 8 [26,27]. A possible effect on splicing can also be speculated for some missense variants, particularly those involving the first or last few nucleotides of each exon (i.e., c.679A > G and c.1102G > C). Indeed, for the c.1102G > C (p.Asp368His), Roosing and colleagues [10] showed that it causes an increased expression of a transcript skipping exon 10.

By whole-genome sequencing, an approximately 2-kb SVA (SINE–VNTR–Alu) retrotransposon insertion in intron 5 was identified by Kim and colleagues [21], producing a missplicing of exon 5 into a cryptic splice acceptor site in *MFSD8* intron 6 and a consequent premature translational termination.

The variant c.863 + 3_4insT was described in three different patients of Italian origin [13] and it was found to be responsible for the synthesis of aberrant transcripts lacking exon 8. The splicing effect of the c.863 + 1G > C variant (described in two patients ([9,13]) and the c.863 + 4A > G variant (one patient [22]) has not been demonstrated in vivo, but only inferred from in silico predictions. In our patient, we demonstrate that the variant c.863 + 2dup prevents the proper synthesis of the full-length transcript, which is the only transcript seen in all brain regions [7]. Apart from the skipping of exon 8, we observed an unexpected transcript also lacking exon 9, indicating that the consequences of a splice variant might be unpredictable. Indeed, considering both the RefSeq release 110 and the GENECODE v.41 databases, no alternative transcripts lacking exon 9 and including exons 7 and 10 are described (i.e., those that can be amplified with the primer pairs we used).

From a clinical point of view, based on reviewing the genetic and clinical data of *CLN7* patients reported in the literature, we found that a quite homogeneous phenotype was described, characterized by developmental regression (89% of patients), motor impairment (88%), seizures or EEG abnormalities (100%), ataxia (68%), cerebral/cerebellar atrophy (80%) and visual impairment (72%). Of note, disease onset occurred between 3 and 5 years of life in most patients (82%) and after 5 years in a few. We also noticed that microcephaly of postnatal onset is another relevant component manifestation of CLN7, affecting about 50% of patients, but head circumference is occasionally reported (17/76 patients, Appendix A). Therefore, periodic evaluation of head circumference is strongly recommended in this condition. Of relevance, our patient showed all classic CLN7 clinical features over time, including progressive postnatal microcephaly and severe visual impairment.

Besides late infantile NCL, biallelic variants in *MFSD8* can be responsible for non-syndromic retinal degeneration (NSRD) of a variable degree, ranging from a milder macular disease [10] to a widespread rod-cone dystrophy with severe macular involvement [11]. NSRD patients are usually compound heterozygous for a missense NSRD-specific allele, with the c.1006C > G (p.Glu33Gln) being the most frequent, and a second allele that may also be responsible for CLN7 [10,11,34,35,36]. A deep intronic variant, c.998 + 1669A > G, was identified with whole-genome sequencing in compound heterozygosity, with the c.929G > A (p.Gly310Asp, CLN7-related) in a patient with isolated cone dystrophy [27]. It was demonstrated to cause an out-of-frame inclusion of a 140 bp pseudoexon from intron 9 (p.Lys333Asnfs*18) and a consequent nonsense-mediated decay. Nonetheless, about 50% of the full-length transcript is still produced from the mutant allele. Only two missense variants were found to be responsible for NSRD in the homozygous state: c.1361T > C (p.Met454Thr) in eight individuals [11,35] and c.1445G > C (p.Arg482Pro) in three siblings [36].

Notably, the splice variant c.750A > G, described above, was identified in compound heterozygosity in two cases: (1) with a frameshift deletion (c.755-2726_998 + 1981delinsGTA) in a patient with the CLN7 classical phenotype; (2) with the NSRD-specific c.1006G > A in a subject with cone-rod dystrophy [27]. It was, instead, homozygous in a pair of siblings with marked clinical differences, since the older brothers displayed signs and symptoms of a juvenile onset neuronal ceroid lipofuscinosis, while the younger sister’s clinical manifestation were only limited to cone-rod dystrophy [26]. 

Kousi and colleagues [9] detected the homozygous missense variant c.468_469delinsCC (p.Ala157Pro) with a milder juvenile onset NCL and a protracted disease course.

A list of *MFSD8* variants reported in the scientific literature as responsible for NCL or isolated retinal degeneration can be found in Table 1.

Considering the described genotype–phenotype correlations, pathogenic *MFSD8* can be classified into two main categories: (1)Null or almost null alleles, leading to a nearly complete loss of protein function;(2)Hypomorphic alleles, with some degree of residual protein function.

The combination of two hypomorphic alleles, or one hypomorphic and one null allele usually cause isolated retinal degeneration, with a variable degree of severity and age of onset, but without central nervous system involvement. On the contrary, the combination of two null alleles leads to the classical late infantile neuronal ceroid lipofuscinosis. Some alleles may display an intermediate behavior with marked phenotypic variability.

Moreover, recently, through case-control NGS association studies, Geier and colleagues suggested that rare variants in *MFSD8* may act as genetic risk factors for frontotemporal dementia (FTLD) [37].

In conclusion, a diagnosis of CLN7 caused by a novel intronic pathogenic variant in *MFSD8* was definitively established in our patient, both genetically, validated by mRNA analysis, and clinically, supported by extensive literature revision. 

Some final considerations are in order. Splicing defects result from a discrete number of causative variants in CLN7, which, in some cases, may not be easily detected by standard diagnostic approaches focusing on coding regions and canonical splice sites. However, their identification and characterization may be crucial not only for diagnostic and counselling purposes, but also because, as already demonstrated [21], at least in some cases, they may constitute the molecular target of a customized oligonucleotide therapy.

Neurodevelopmental disorders (NDDs) define a group of conditions characterized by extreme clinical and genetic heterogeneity. NDDs can appear to be associated with distinctive dysmorphic features, allowing the genetic diagnosis to be raised clinically in several cases, or they can occur as an isolated feature or associated with nonspecific multisystem clinical signs. For all these reasons, genome-wide genotyping is increasingly applied as a first-tier test for the genetic diagnosis of unexplained NDDs. However, this “genotype-first” approach generates large amounts of genomic data that can prevent consistent diagnoses, especially if it is not connected with critical clinical assessment.

In our case, exome sequencing (ES) would have detected the intronic splice site variant. However, we consider that further investigations at the mRNA level, for validation of variants of uncertain significance [38], are not obvious during a routine application of ES. Our case is discussed to highlight the pivotal role of clinical genetics, and of a multidisciplinary evaluation, in driving a linear, unequivocal and not time-consuming diagnostic procedure.

## Figures and Tables

**Figure 1 genes-14-00245-f001:**
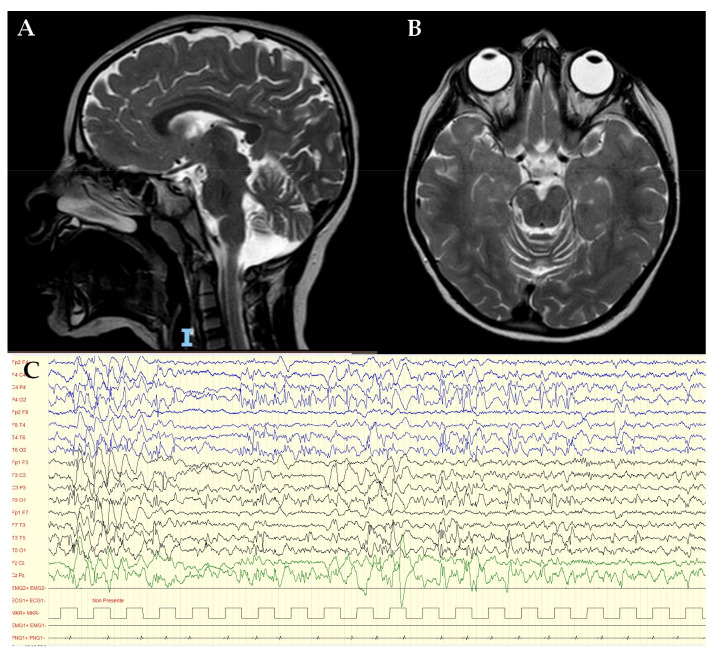
Sagittal (**A**) and axial (**B**) brain and spinal cord MRI showed diffuse and bilateral cerebellar atrophy and bilateral and symmetrical hyperintensity of the posterior periventricular white matter and mesial portions of the thalami, on T2 and FLAIR sequences; (**C**) EEG show focal epileptiform anomalies in the middle-posterior regions and diffuse anomalies.

**Figure 2 genes-14-00245-f002:**
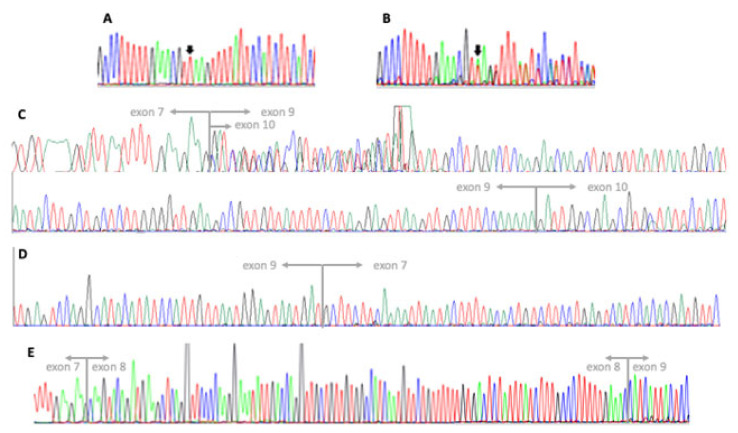
Electropherograms of sequencing analyses: (**A**) exon 8–intron 8 junction of the *MFSD8* gene (NM_001371596.2, forward strand) in our patient, presenting with a homozygous insertion of a thymine between the 2nd and 3rd position of the donor splice site (c.863 + 2dup); (**B**) exon 8–intron 8 junction of the *MFDS8* gene (forward strand) in the patient’s mother showing the heterozygous variant; (**C**) *MFSD8* cDNA analysis in our patient showing two overlapping sequences both skipping exon 8, with the shorter lacking exon 9 as well. Junctions between exons 7–9 (7–10 in the shorter fragment) and 9–10 are indicated; (**D**) MFSD8 cDNA analysis in our patient displaying the junction between exons 7 and 9 on the reverse strand; (**E**) MFSD8 cDNA analysis in a control subject displaying both the junctions between exons 7–8 and 8–9.

**Table 1 genes-14-00245-t001:** Pathogenic *MFSD8* variants reported in the scientific literature (CLN7 and NSRD).

Nucleotide Change [NM_001371596.2]	Protein Change [NP_001358525.1]	[Reference]: n. of Patients- zygosity	LINCL	NS- RD *	Other
c.2T > C	p.?	[13]: 1 ComHet	1		
c.103C > T	p.(Arg35 *)	[9]: 1 Homo, [11]: 1 ComHet, [6]: 1 Homo + 1 ComHet, [13]: 1 Homo	4	1	
c.[104G > A;155G > C]	p.[(Arg35Gln);(Gly52Ala)]	[27]: 1 ComHet		1	
c.136_137del	p.(Met46Valfs * 22)	[30]: 1 ComHet	1		
c.154G > A	p.(Gly52Arg)	[13]: 1 ComHet [34]: 1 ComHet	1	1	
c.233G > A	p.(Trp78 *)	[11]: 2 ComHet		2 (2)	
c.259C > T	p.(Gln87 *)	[6]: 1 Homo	1		
c.301G > C	p.(Ala146Pro)	[24]: 1 Homo	1		
c.325_339del	p.(Val109_Ile113del)	[19]: 1 Homo	1		
c.362A > G	p.(Tyr121Cys)	[14]: 5 Homo	5 (5)		
c.416G > A	p.(Arg139His)	[9]: 1 Homo, [22]: 1 Homo, [6]: 1 Homo	3		
c.468_469delinsCC	p.(Ala157Pro)	[9]: 1 Homo			jNCL
c.472G > A	p.(Gly158Ser)	[16]: 5 Homo	5 (5)		
c.479C > A	p.(Thr160Asn)	[6]: 1 Homo	1		
c.479C > T	p.(Thr160Ile)	[6]: 1 ComHet	1		
c.525T > A	p.(Cys175*)	[18]: 1 Homo	1		
c.554-1G > C	p.?	[6]: 1 ComHet	1		
c.554-5A > G	p.?	[23]: 1 ComHet	1		
SVA insertion	p.[Val185Aspfs * 3,?]	[21]: 1 ComHet	1		
c.557T > G	p.(Phe186Cys)	[28]: 1 ComHet	1		
c.627_643del	p.(Met209Ilefs * 3)	[9]: 1 Homo, [13]: 1 Homo	1		
c.679A > G	p.(Arg233Gly)	[12]: 1 Homo	1		
c.721G > T	p.(Gly241 *)	[24]: 1 Homo	1		
c.750A > G	p.[Arg233Serfs * 5,=]	[26]: 2 Homo, [27]: 2 ComHet	1	1	2 jNCL
c.754 + 1G > A	p.?	[6]: 1 Homo, 1 ComHet	2		
c.754 + 2T > A	p.[Arg233Serfs * 5,=]	[6]: 6 Homo 2 + ComHet, [7]: 1 Homo, [12]: 1 Homo, [20]: 1ComHet, [22]: 1 Homo	12 (2)		
c.755- 2726_998 + 1981delinsGTA	p.(Ala252Glyfs * 82)	[27]: 1 ComHet	1		
c.850G > C	p.(Ala284Pro)	[25]: 1 Homo	1		
c.863 + 1G > C	p.?	[9]: 1 Homo, [13]: 1 Homo	2		
c.863 + 3_863 + 4insT	p.?	[13]: 3 ComHet	3		
c.863 + 4A > G	p.?	[22]: 1 Homo	1		
c.881C > A	p.(Thr294Lys)	[9]: 18 Homo, [6]: 3 Homo + 1 ComHet, [13]: 1 ComHet, [20]: 1 ComHet, [22]: 1 Homo, [27]: 2 ComHet	25 (2,2,2)	2	
c.894T > G	p.(Tyr298 *)	[7]: 1 Homo	1		
c.929G > A	p.(Gly310Asp)	[7]: 1 Homo, [13]: 1 ComHet, [6]: 1 Homo, [27]: 1 ComHet	3	1	
c.998 + 1669A > G	p.[=,Lys333Asnfs * 18]	[27]: 1 ComHet		1	
c.1006C > G	p.(Glu336Gln)	[10]: 6 ComHet, [11]: 6 ComHet, [34]: 1 ComHet, [27]: 2 ComHet		13 (5,3,2)	
c.1006G > A	p.(Glu336Lys)	[27]: 1 ComHet		1	
c.1009C > T	p.(Arg337Cys)	[27]: 1 ComHet		1	
c.1093C > T	p.(Gln365 *)	[24]: 2 Homo	2		
c.1102G > C	p.[Asp368His, Ile334Phefs * 4]	[7]: 1 Homo, [21]: 1ComHet, [10]: 1 ComHet	2	1	
c.1103-2del	p.?	[9]: 1 ComHet	1		
c.1141G > T	p.(Glu381 *)	[13]: 1 ComHet, [10]: 5 ComHet, [27]: 1 ComHet	1	6 (5)	
c.1219T > C	p.(Trp407Arg)	[17]: 3 ComHet	3		
c.1235C > T	p.(Pro412Leu)	[15]: 3 Homo, [6]: 1 ComHet, [35]: 3 ComHet	4 (3)	3	
c.1241_1242ins GAAT	p.(Ile414Metfs * 109)	[22]: 1 Homo	1		
c.1265C > A	p.(Ser422 *)	[27]: 1 ComHet		1	
c.1286G > A	p.(Gly429Asp)	[7]: 1 Homo	1		
c.1340C > T	p.(Pro447Leu)	[13]: 1 Homo	1		
c.1351-1G > A	p.?	[28]: 1 ComHet	1		
c.1361T > C	p.(Met454Thr)	[17]: 3 ComHet, [11]: 6 Homo, [35]: 3 ComHet, 2 Homo	3	11 (3,2)	
c.1373C > A	p.(Thr458Lys)	[6]: 1 ComHet	1		
c.1393C > T	p.(Arg465Trp)	[9]: 1Homo	1		
c.1394G > A	p.(Arg465Gln)	[6]: 1 ComHet, [22]: 1 Homo, [11]: 3 ComHet	2	3 (3)	
c.1408A > G	p.(Met470Val)	[6]: 1 ComHet	1		
c.1420C > T	p.(Gln474 *)	[6]: 1 Homo	1		
c.1444C > T	p.(Arg482 *)	[13]: 1 ComHet, [6]: 1 ComHet, [23]: 1 ComHet,	3		
c.1445G > C	p.(Arg482Pro)	[36]: 3 Homo		3 (3)	
Deletion exon1–4	p.?	[29]: 1 Homo	1		
Whole gene deletion (180 kb)		[30]: 1 ComHet	1		

LINCL: late infantile NCL; NSRD: non-syndromic retinal dystrophy; jNCL: juvenile NCL; Homo: homozygous; ComHet: compound heterozygous; * in parentheses nr. of related individuals from different families.

## Data Availability

Data are contained within the article and Appendix A.

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
