# Peer review of "Linear Diagnostic Procedure Elicited by Clinical Genetics and Validated by mRNA Analysis in Neuronal Ceroid Lipofuscinosis 7 Associated with a Novel Non-Canonical Splice Site Variant in MFSD8"

_genes, 2023, doi:10.3390/genes14020245_

Round 1

Reviewer 1 Report

Line 39: Loss of retinal neurons is not a universal feature of the NCLs, so this should not be stated here.

Line 41: "proteins-coding ..." should be "protein-coding ..."

Line 41: Delete the word "lysosomal" since not all transmembrane NCL proteins have been shown to localize to lysosomes exclusively if at all.

Line 42: "proteins-coding ..." should be "protein-coding ..."

Lines 45-47  Change to "It is a 518 amino acid lysosomal transmembrane protein that appears to be involved in the translocation of small solutes across the membrane."

Line 50: Delete the word "Recently" as 2015 is not considered recent.

Lines 52-55: Cite the UCL NCL database that lists known human MFSD8 variants associated with NCL CLN7 / MFSD8 | NCL Disease - UCL – University College London

Clinical assessment of patient:  Nothing is said about an ophthalmological exam or assessment of visual signs in the subject patient.  Since these signs are a key component of the disease in the majority of CLN7 patients, this is an important omission that should be addressed.

Line 93: First word should be "heparinzed"

Line 102: Need to provide information on how the "control" subject was selected and whether the control sample was also from lymphocyte cultures.

Lines 197-198:  The authors are not justified in saying anything about the accumulation of ceroid-lipopigment, since they did not analyze for this.

Discussion starting on line 202:  The authors need to reference the UCL NCL database cited above and indicate what additional information they are adding.  

Line 229:  The authors indicate that of the cases they reviewed, 68% reported visual impairment.  This reinforces the earlier comment that the child evaluated in this study should have been examined for signs of visual impairment and retinal degeneration.

Author Response

We thank very much the reviewer for the valuable time reviewing our work and the insightful comments that helped to improve the quality of the paper.

Line 39: Loss of retinal neurons is not a universal feature of the NCLs, so this should not be stated here.

We specified that retina is involved in most (not all) NCL forms.

Line 41: "proteins-coding ..." should be "protein-coding ..."

Line 41: Delete the word "lysosomal" since not all transmembrane NCL proteins have been shown to localize to lysosomes exclusively if at all.

Line 42: "proteins-coding ..." should be "protein-coding ..."

Lines 45-47  Change to "It is a 518 amino acid lysosomal transmembrane protein that appears to be involved in the translocation of small solutes across the membrane."

Line 50: Delete the word "Recently" as 2015 is not considered recent.

We made all the corrections as suggested.

Lines 52-55: Cite the UCL NCL database that lists known human MFSD8 variants associated with NCL CLN7 / MFSD8 | NCL Disease - UCL – University College London

We thank very much the reviewer for the precious suggestion. We added the proposed citation.

Clinical assessment of patient:  Nothing is said about an ophthalmological exam or assessment of visual signs in the subject patient.  Since these signs are a key component of the disease in the majority of CLN7 patients, this is an important omission that should be addressed.

We have now specified that a first ophthalmologic evaluation revealed a significantly impaired visual acuity. We suggested a more in-depth investigation, with an assessment of the ocular fundus and OCT, after the molecular diagnosis was reached and, by this time, the patient has not yet undergone the proposed evaluation.

Line 93: First word should be "heparinzed"

We made the correction as suggested.

Line 102: Need to provide information on how the "control" subject was selected and whether the control sample was also from lymphocyte cultures.

We clarified that the control RNA sample was obtained from a healthy subject using the same procedure.

Lines 197-198:  The authors are not justified in saying anything about the accumulation of ceroid-lipopigment, since they did not analyze for this.

We rephrased the sentence to avoid that statement.

Discussion starting on line 202:  The authors need to reference the UCL NCL database cited above and indicate what additional information they are adding.  

We cited the UCL NCL database. After a comparison, we recalculated the percentages of variant types, and we specified the criteria we used for the inclusion of the reported patients among those considered in our tables (neither personal communications, nor cases with isolated retinal involvement).

We did not aim to add further information to the database, apart from the patient we are describing and the study we performed to reach a proper evaluation of the variant. We just wanted to summarize some clinical and genetic aspects from both literature and database mining. However, we cited  further articles that have not yet been included in the UCL NCL database, but we think there is no need to highlight it in the manuscript.

Line 229:  The authors indicate that of the cases they reviewed, 68% reported visual impairment.  This reinforces the earlier comment that the child evaluated in this study should have been examined for signs of visual impairment and retinal degeneration.

We added the results of a preliminary ophthalmologic evaluation, as stated above.

Reviewer 2 Report

Authors detected a novel homozygous non-canonical splice-site variant in MFSD8 in a 5-year-old girl who had progressive neurocognitive impairment and microcephaly by clinical genetic test first then confirmed by cDNA Seq. and brain imaging. Authors showed that 3 AR genes lying within the 24 Mb regions of homozygosity were consistent with the clinical phenotype including EXOSC9, SPATA5, and MFSD8. They also performed MFSD8 Seq. and showed that there was a skipping of exon 8 and they defined this variant as pathogenic.

Authors need to include all references supporting the information mentioned in different sections of the manuscript as indicated below.

Introduction

Line 52: These sentences need references “Since the initial identification of the causative gene in 2007, more than 50 different MFSD8 variants have been reported in literature. Most patients were diagnosed with missense and stop-gain variants; however, canonical splice site variants, single exon or whole exon deletions were also detected in a few”.

Materials and methods

Line 71: This sentence needs references as well “Clinical and genetic data of our patient were comparatively evaluated with those of patients reported in the literature”.

For “Analyses on genomic DNA”, please provide reference/s if these analyses were published before.

Line 81: I suggest adding the primer sequences as supplementary materials and methods.

Line 92: Please provide the references for “Short-term PHA-stimulated lymphocyte cultures were established from patient’s eparinated blood samples, following standard procedures. Total RNA was hence extracted from both untreated and puromycin-treated cells using the guanidinium thiocyanate-phenol-chloroform extraction method”.

Line 93: Please correct “heparinated” blood samples.

Results

Figure 1: Please correct “(B)” instead of axial (A) brain.

Line 157: Authors mentioned that “To investigate the possible consequences of the variant at mRNA level, we performed cDNA sequencing”, can they explain what is meant by the variant at mRNA level? And can they better clarify what the advantage of adding the cDNA Seq. to genomic Seq. in this study is? Because cDNA is a DNA copy of mRNA which is a transcript of genomic DNA sequenced first. Why did the authors expect to see a difference?

Discussion

Line 209: Please correct “introns” instead of intron.

Line 202: Please move this sentence “We performed a revision of genetic and clinical data of CLN7 patients reported in the literature (tot 71, see Supplements)[9–20].“ to the Materials and Methods section (Line 72).

In the discussion section, authors have to discuss their findings relative to literature, so I suggest to add something like “Based on reviewing the genetic and clinical data of CLN7 patients reported in the literature, we found that…..”.

The manuscript needs minor language editing.

Author Response

We thank very much the reviewer for the valuable time reviewing our work and the insightful comments that helped to improve the quality of the paper.

Authors detected a novel homozygous non-canonical splice-site variant in MFSD8 in a 5-year-old girl who had progressive neurocognitive impairment and microcephaly by clinical genetic test first then confirmed by cDNA Seq. and brain imaging. Authors showed that 3 AR genes lying within the 24 Mb regions of homozygosity were consistent with the clinical phenotype including EXOSC9SPATA5, and MFSD8. They also performed MFSD8 Seq. and showed that there was a skipping of exon 8 and they defined this variant as pathogenic.

Authors need to include all references supporting the information mentioned in different sections of the manuscript as indicated below.

Introduction

Line 52: These sentences need references “Since the initial identification of the causative gene in 2007, more than 50 different MFSD8 variants have been reported in literature. Most patients were diagnosed with missense and stop-gain variants; however, canonical splice site variants, single exon or whole exon deletions were also detected in a few”.

We added all references as suggested.

Materials and methods

Line 71: This sentence needs references as well “Clinical and genetic data of our patient were comparatively evaluated with those of patients reported in the literature”.

We added all references as suggested.

For “Analyses on genomic DNA”, please provide reference/s if these analyses were published before.

Described methods (i.e.: CGH+SNP Microarray and Sanger sequencing) were performed following manufacturer instructions, as stated. Since the main references were just the manuals provided together with the reagents we used, we do not have any article to cite detailing these procedures.   

Line 81: I suggest adding the primer sequences as supplementary materials and methods.

We added primer sequences in the Supplementary Table S3.

Line 92: Please provide the references for “Short-term PHA-stimulated lymphocyte cultures were established from patient’s eparinated blood samples, following standard procedures. Total RNA was hence extracted from both untreated and puromycin-treated cells using the guanidinium thiocyanate-phenol-chloroform extraction method”.

We added references for both short-term PHA-stimulated lymphocyte cultures and RNA extraction.

Line 93: Please correct “heparinated” blood samples.

We made the corrections as suggested.

Results

Figure 1: Please correct “(B)” instead of axial (A) brain.

We made the corrections as suggested.

Line 157: Authors mentioned that “To investigate the possible consequences of the variant at mRNA level, we performed cDNA sequencing”, can they explain what is meant by the variant at mRNA level? And can they better clarify what the advantage of adding the cDNA Seq. to genomic Seq. in this study is? Because cDNA is a DNA copy of mRNA which is a transcript of genomic DNA sequenced first. Why did the authors expect to see a difference?

We reworded the sentence specifying what we expected to find by sequencing the cDNA that cannot be seen with the sequencing of genomic DNA, since the variant was intronic.

Discussion

Line 209: Please correct “introns” instead of intron.

Line 202: Please move this sentence “We performed a revision of genetic and clinical data of CLN7 patients reported in the literature (tot 71, see Supplements)[9–20].“ to the Materials and Methods section (Line 72).

In the discussion section, authors have to discuss their findings relative to literature, so I suggest to add something like “Based on reviewing the genetic and clinical data of CLN7 patients reported in the literature, we found that…..”.

The manuscript needs minor language editing.

We made the corrections as suggested.